# Photoactivation of silicon rhodamines via a light-induced protonation

Michelle S. Frei [1,2], Philipp Hoess [3,4], Marko Lampe [5], Bianca Nijmeijer [3], Moritz Kueblbeck[3], Jan Ellenberg [3], Hubert Wadepohl[6], Jonas Ries [3], Stefan Pitsch[7], Luc Reymond [8,9]* & Kai Johnsson[1,2,9]*

Photoactivatable fluorophores are important for single-particle tracking and super-resolution microscopy. Here we present a photoactivatable fluorophore that forms a bright silicon rhodamine derivative through a light-dependent protonation. In contrast to other photoactivatable fluorophores, no caging groups are required, nor are there any undesired side-products released. Using this photoactivatable fluorophore, we create probes for HaloTag and actin for live-cell single-molecule localization microscopy and single-particle tracking experiments. The unusual mechanism of photoactivation and the fluorophore's outstanding spectroscopic properties make it a powerful tool for live-cell super-resolution microscopy.

[1] Department of Chemical Biology, Max Planck Institute for Medical Research, Jahnstrasse 29, 69120 Heidelberg, Germany. [2] Institute of Chemical Sciences and Engineering (ISIC), École Polytechnique Fédérale de Lausanne (EPFL), 1015 Lausanne, Switzerland. [3] Cell Biology and Biophysics Unit, European Molecular Biology Laboratory (EMBL), Meyerhofstrasse 1, 69117 Heidelberg, Germany. [4] Collaboration for joint PhD degree between EMBL and Heidelberg University, Faculty of Biosciences, Heidelberg, Germany. [5] Advanced Light Microscopy Facility (ALMF), European Molecular Biology Laboratory (EMBL), Meyerhofstrasse 1, 69117 Heidelberg, Germany. [6] Anorganisch-Chemisches Institut, University of Heidelberg, Im Neuenheimer Feld 270, 69120 Heidelberg, Germany. [7] Spirochrome AG, Chalberweidstrasse 4, CH-8260 Stein am Rhein, Switzerland. [8] Biomolecular Screening Facility, École Polytechnique Fédérale de Lausanne (EPFL), 1015 Lausanne, Switzerland. [9] National Centre of Competence in Research (NCCR) in Chemical Biology, 1015 Lausanne, Switzerland. *email: luc.reymond@epfl.ch; johnsson@mr.mpg.de

Advances in super-resolution microscopy (SRM) have led to insights into cellular structures and processes over the past decade[1,2]. One of these SRM approaches is single-molecule localization microscopy (SMLM), which relies on the switching of fluorophores between an off and an on state[3–5]. The switching can be achieved by using photoactivatable or switchable fluorophores[6–8]. Up to date, fluorescent proteins and small-molecule fluorophores are the two most commonly used fluorophore classes in SMLM. For live-cell experiments with intracellular proteins, fluorescent proteins have the main advantage that no chemical labeling steps are required. However, small-molecule fluorophores are generally brighter and more photostable than fluorescent proteins[9], and therefore of advantage for SMLM experiments[10,11]. Photoactivatable (or caged) small-molecule fluorophores are known throughout many of the different fluorophore families and are mainly synthesized using photolabile protecting groups[11–14]. Photoactivatable rhodamine derivatives have been obtained through the attachment of *ortho*-nitrobenzyl moieties[11]. However, these probes are mostly used in fixed-cell microscopy due to their decreased solubility and poor cell-permeability[15–17]. Furthermore, they result in the stoichiometric formation of very electrophilic nitroso-aldehydes or ketones as reactive byproducts, which are toxic and of concern in live-cell imaging[18]. Rhodamines have also been rendered photoactivatable through a diazoketone group[19], leading to the introduction of the photoactivatable Janelia Fluor dyes PA-JF$_{549}$ and PA-JF$_{646}$[20], which have been successfully used for fixed-cell and live-cell SMLM. However, photoactivation of these fluorophores leads to the formation of a dark side-product. The extent, to which the undesired side-product is formed, depends on the structure and environment of the fluorophore complicating applications of the diazoketone approach. In addition, photoactivation of fluorophores caged with the diazoketone group proceeds through a carbene, which can react with intracellular nucleophiles (Supplementary Fig. 1)[21]. In light of the limitations of the existing caging strategies, alternative chemical strategies are needed to generate photoactivatable fluorophores.

Here, we report the discovery, synthesis and characterization of a class of cell-permeable, photoactivatable fluorophores (PA-SiRs), which are based on the silicon rhodamine (SiR) scaffold and activated through a light-induced protonation. We demonstrate the utility of these fluorophores for live-cell SMLM of intracellular targets and single-particle tracking experiments.

## Results

**Synthesis and characterization**. The first analog of this class of fluorophores was serendipitously found during the attempted synthesis of a SiR derivative bearing an alkyl chain in place of the aromatic substituent at the 9 position of the xanthene scaffold (Fig. 1a). Instead of the desired fluorescent SiR **2** we isolated the non-fluorescent analog PA-SiR (**1**) (Supplementary Fig. 2). PA-SiR possesses an exocyclic double bond and the two aromatic ring systems are not conjugated, reflected by its $\lambda_{abs,max}$ value of 290 nm and demonstrated by the X-ray crystal structure of PA-SiR analog **4** (Fig. 1b). However, PA-SiR underwent protonation upon ultraviolet (UV) irradiation in aqueous solution, re-establishing the fluorescent xanthene core of SiR **2** (Fig. 1a, c and Supplementary Figs. 3 and 4). The photoproduct SiR **2** showed an absorption maximum at $\lambda_{abs,max} = 646$ nm and emitted at around 660–670 nm. Its extinction coefficient of $\varepsilon_{646} = 90,000 \pm 18,000$ M$^{-1}$ cm$^{-1}$ and fluorescence quantum yield $\varphi = 19.0 \pm 2.4\%$ in aqueous buffer (mean ± 95% confidence interval, $N = 3$ and 4 samples, respectively) were only marginally smaller than those of the previously described SiR-carboxyl[22] (Fig. 1c and Supplementary Table 1). However, **2** is susceptible to nucleophilic

attack by water leading to rapid establishment of an equilibrium between **2** and **3** (Fig. 1a, d, f and Supplementary Fig. 5). Structural modifications on PA-SiR can influence this equilibrium as demonstrated by several synthetized analogs (Supplementary Fig. 6). Moreover, both photoactivation of PA-SiR as well as the equilibrium between **2** and **3** are pH sensitive (Fig. 1e and Supplementary Fig. 7). Photoactivation is prevented by protonation of the aniline groups and is therefore highest at pH values above pH = 6 as revealed by measuring the maximal absorbance at 646 nm reached directly after activation ($A_{max}$). The equilibrium between **2** and **3**, as measured by recording the absorbance at equilibrium and correcting for $A_{max}$ at 646 nm ($A_{eq}$), was shifted toward **3** at higher pH values (Fig. 1e). At physiological pH only about 10% of the activated PA-SiR was present as SiR **2** in comparison to 80% at pH = 6.1. Noteworthy is also the quantitative nature of the photoconversion of PA-SiR, which becomes apparent when following the conversion of PA-SiR to **3** by nuclear magnetic resonance (NMR). These experiments also revealed that the photoactivation is reversible on a time scale of days (Fig. 1f and Supplementary Fig. 5).

Calculations of the frontier molecular orbitals of model compound PA-SiR **4** and data published on cross-conjugated 1,1-diphenyl alkenes[23,24] indicate that the photoactivation could proceed through a twisted intramolecular charge transfer followed by protonation of the intermediate (Supplementary Fig. 3). Further studies are needed to clarify the mechanism of this remarkable reaction and to the best of our knowledge this type of light-induced protonation has not previously been reported for rhodamine derivatives or other xanthenes. Carbopyronine and fluorescein derivatives with such an exocyclic double bond have been described (see Supplementary Fig. 3 for an overview of related structures and reactions), but have not been reported to undergo light-induced protonation. However, with the many strategies available to tune the HOMOs and LUMOs of xanthene derivatives, we believe it should be feasible to generate photoactivatable rhodamine and carbopyronine derivatives, thereby expanding the scope of this reaction.

The susceptibility of activated PA-SiR toward nucleophiles and its half-life of minutes at physiological pH are a disadvantage of these fluorophores for standard diffraction limited imaging. However, this is less relevant for single-molecule based super-resolution microscopy since the observation period of individual fluorophores in SMLM is on the order of milliseconds and the reaction of activated PA-SiR with nucleophiles should not interfere in such experiments. Furthermore, the equilibrium of the reaction of activated PA-SiR with nucleophiles is environmentally sensitive. In fact, when we prepared conjugates of PA-SiR with ligands for protein labeling (Supplementary Figs. 8 and 10)[25–28], we discovered that PA-SiR-Halo attached to HaloTag (Supplementary Fig. 9) can be very efficiently activated and its fluorescent form **2** is stable over hours at physiological pH, whereas PA-SiR-Halo not conjugated to HaloTag is inefficiently activated and the activated probe decays quickly (Fig. 2a, b, Supplementary Table 1 and Supplementary Fig. 10). We attribute the remarkable stability of PA-SiR-Halo attached to HaloTag to specific interactions between HaloTag and the activated probe. It has been previously observed that HaloTag possesses a high affinity toward zwitterionic rhodamine derivatives, including SiR-carboxyl[22]. However, in the absence of structural information on HaloTag labeled with PA-SiR-Halo we cannot provide more detailed insights on the nature of these interactions. This apparent fluorogenicity of the probe should prove beneficial for live-cell imaging as unconjugated PA-SiR-Halo is not fluorescent, which increases the signal-to-background ratio. In addition, PA-SiR-Halo conjugated to HaloTag and

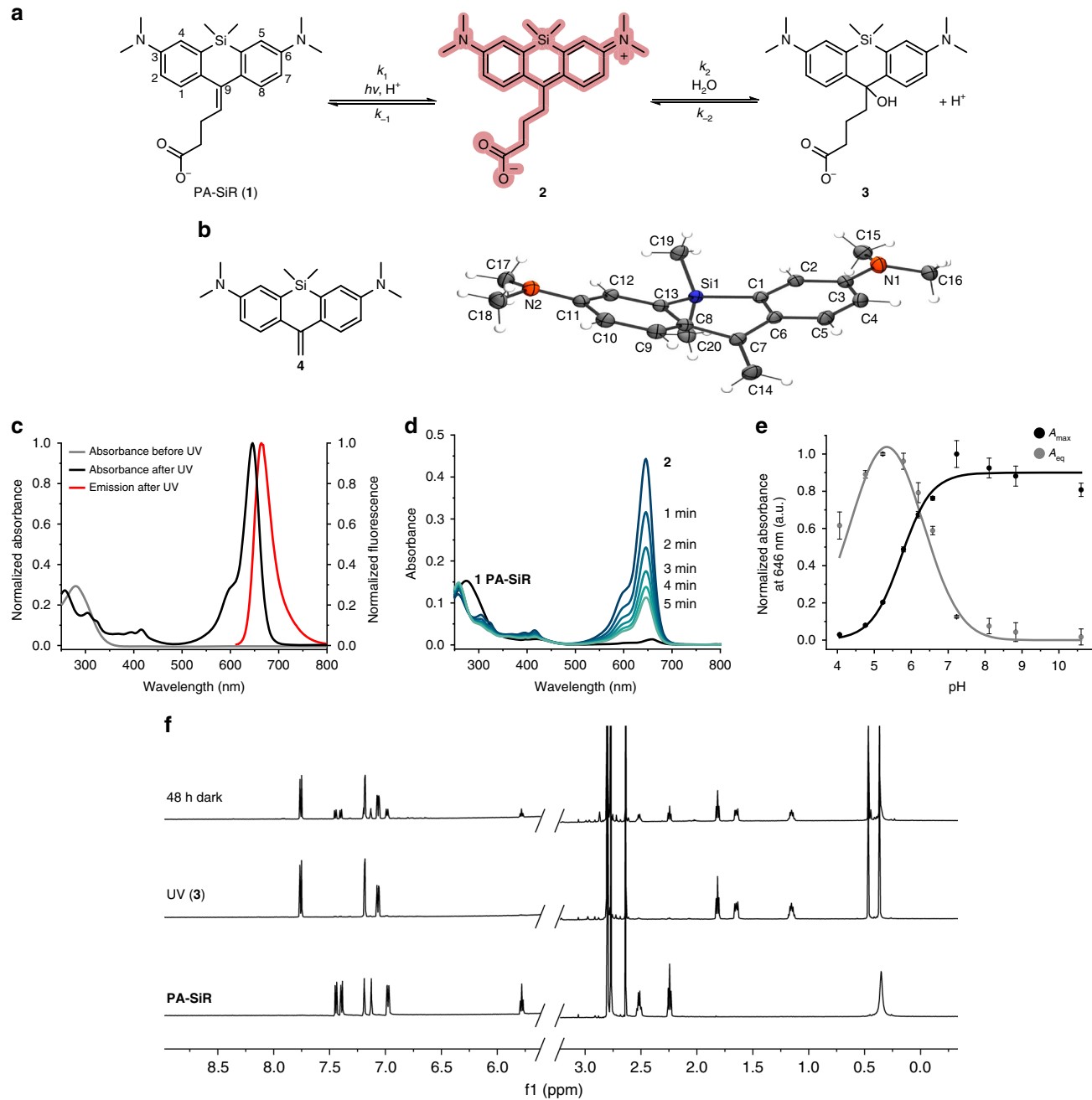

**Fig. 1** Structure and properties of PA-SiR. **a** Reaction scheme for photoactivation of PA-SiR (**1**), and equilibrium between **2** and **3**. **b** Chemical structure of compound **4** together with its Oak Ridge Thermal Ellipsoid Plot (ORTEP), arbitrary numbering. Atomic displacement parameters are drawn at 50% probability level. Selected bond lengths (Å) and torsion angles (°): C4–C5 1.380(2), C5–C6 1.3999(19), C6–C7 1.4896(19), C7–C8 1.4929(19), C7–C14 1.344(2), C1–C6–C7–C14 145.94(15), C5–C6–C7–C14 −31.9(2), C7–C8–C13–Si1 −2.91(17), C14–C7–C8–C9 33.4(2). The C7–C14 bond is the shortest bond followed by the aromatic bonds exemplified by C4–C5 or C5–C6. Bonds C6–C7 and C7–C8 are considerably longer. Further information can be found in Supplementary Table 11. **c** Normalized absorption spectra of PA-SiR in PBS (10 µM) before and after UV irradiation as well as emission spectra after activation. **d** Absorption spectra of PA-SiR in PBS (10 µM) before activation and directly after UV irradiation measured every 1 min, revealing the reaction from **2** to **3**. **e** pH dependence of the equilibrium system of PA-SiR in PBS (10 µM) at different pH after brief photoactivation through UV irradiation. Normalized absorbance values $A_{max}$ directly after activation and $A_{eq}$ in equilibrium at different pH values are given, reflecting changes in activation ($A_{max}$) and equilibrium constant ($A_{eq}$). Values displayed are means from three individual measurements, error bars correspond to 95% confidence intervals. Source data are provided as a Source Data file. **f** $^1$H nuclear magnetic resonance (NMR) spectra of PA-SiR (2.0 mM in PBS) before UV irradiation, after complete conversion to **3** and after further 48 h in the dark. (For assignment of peaks see Supplementary Fig. 5)

photoactivated showed much greater stability toward other nucleophiles such as cysteamine than free PA-SiR (Fig. 2c). The generated fluorescent product had an extinction coefficient of $\varepsilon_{646} = 180{,}000 \pm 30{,}000\ \text{M}^{-1}\ \text{cm}^{-1}$ and a fluorescence quantum yield of $\varphi = 29.2 \pm 1.2\%$ in aqueous buffer (mean ± 95%

confidence interval, $N = 3$ samples) making it an outstanding fluorophore. Its quantum yield of activation was found to be $\varphi_{act} = 0.86 \pm 0.07\%$ at 340 nm and $\varphi_{act} = 0.09 \pm 0.04\%$ at 405 nm (mean ± standard error of the mean, $N = 3$ samples), similar to that of PA-JF$_{646}$ (Supplementary Tables 1 and 2)[29].

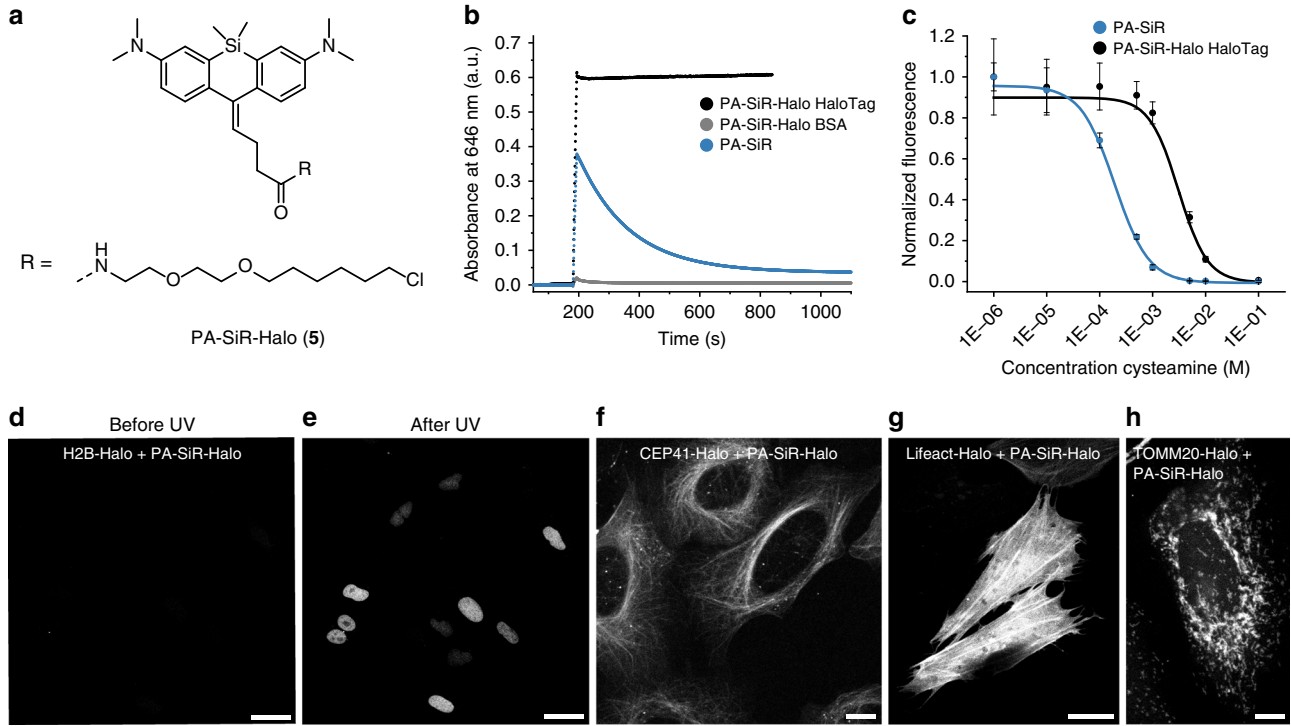

**Fig. 2** PA-SiR-Halo and the influence of HaloTag on its equilibrium system. **a** Chemical structure of PA-SiR-Halo (**5**). **b** Absorbance measurements at 646 nm over time for PA-SiR and PA-SiR-Halo in PBS (10 μM). PA-SiR-Halo was measured with addition of BSA or HaloTag (20 μM). **c** Fluorescence signal after addition of cysteamine (0.001–100 mM) to fully activated PA-SiR or PA-SiR-Halo on HaloTag solutions in equilibrium (1 μM dye on 2 μM HaloTag). The effective concentrations at which half maximal fluorescence intensity was reached (EC$_{50}$ values) were determined to be 0.192 ± 0.019 mM for PA-SiR and 3.1 ± 0.5 mM for PA-SiR-Halo (mean ± 95% confidence interval, both $N = 24$ samples), error bars correspond to 95% confidence intervals. Source data are provided as a Source Data file. **d**, **e** Maximum projection of a z-stack of U-2 OS cells stably expressing H2B-Halo (nucleus) stained with PA-SiR-Halo (0.5 μM for 2 h) before **d** and after UV irradiation **e** Scale bar, 40 μm. **f–h** Confocal images of several HaloTag fusion proteins stained with PA-SiR-Halo (0.5 μM for 1.5 h): **f** microtubules (CEP41-Halo). Scale bar, 10 μm. **g** F-actin (LifeAct-Halo). Scale bar, 20 μm. **h** the outer mitochondrial membrane (TOMM20-Halo). Scale bar, 10 μm

**Live-cell confocal imaging**. PA-SiR-Halo possesses a number of properties that make it an attractive candidate for live-cell imaging such as the absence of side-products during photoconversion, the absence of caging groups that affect solubility and permeability, the efficiency of photoactivation and stability of the HaloTag-bound probe compared to unconjugated probe, and its outstanding spectroscopic properties. We, therefore, incubated U-2 OS cells expressing a histone H2B-HaloTag fusion protein with 0.5 μM PA-SiR-Halo for 2 h and imaged the cells prior and after UV activation at 365 nm (Fig. 2d, e). We found that PA-SiR-Halo showed an excellent signal-to-background ratio after activation under no wash conditions (32 ± 5, mean ± 95% confidence interval, $N = 119$ cells) and that the fluorescence signal after activation was stable over time. In comparison, PA-JF$_{646}$-Halo showed faster activation kinetics but a lower signal-to-background ratio after activation (13.2 ± 1.9, $N = 121$ cells) (Supplementary Fig. 12d–f)[20]. Moreover, PA-SiR-Halo was used to image various other intracellular HaloTag fusion proteins (Fig. 2f–h). Taken together, these experiments validate that PA-SiR-Halo is suitable for live-cell imaging. It should be noted that other PA-SiR probes can be generated (Supplementary Figs. 8 and 10). Specifically, we attached PA-SiR to the F-actin-binding natural product jasplakinolide, yielding PA-SiR-actin, and used it successfully for live-cell imaging of actin filaments (Supplementary Figs. 10b and 12c)[27,28].

**Fixed-cell SMLM**. The photophysical properties such as the number of detected photons per frame and fluorophore are

decisive for SMLM as the attainable localization precision scales with the inverse square root of the number of detected photons[30]. In order to determine these numbers, we immobilized HaloTag labeled with PA-SiR-Halo on coated glass coverslips and imaged the fluorophore using total-internal reflection (TIRF) microscopy (Supplementary Fig. 13a). In these experiments, we used a 405 nm laser for photoactivation, generally used to create a sparse subset of fluorescent molecules in SMLM. We found that the photon numbers per particle per frame for PA-SiR-Halo at a power density of 1.2 kW cm$^{-2}$ suitable for live-cell single-particle tracking were roughly 30% higher than for PA-JF$_{646}$-Halo and considerably higher than those measured for mEos3.2 (Supplementary Fig. 13b)[20]. To test the performance of PA-SiR-Halo in fixed-cell SMLM, we expressed the microtubule binding protein Cep41 as a HaloTag fusion in U-2 OS cells and labeled it with PA-SiR-Halo. The microtubule diameter was determined to be FWHM$_{PA-SiR-Halo}$ = 38.7 ± 7.7 nm (mean ± 95% confidence interval, $N = 20$ tubules) which corresponds well to the microtubule diameter of 25 nm if one takes the size of Cep41-Halo (74 kDa, ca. 5 nm) into account. Moreover, the data is consistent with previously reported data (Fig. 3a, Supplementary Fig. 14a, g)[8,31].

Furthermore, we imaged a HaloTag fusion of Nup96[32], a protein of the nuclear pore complex. Nuclear pores possess a regular circular shape with an internal diameter of about 100 nm[33,34]. Using PA-SiR-Halo labeled Nup96-Halo in fixed U-2 OS cells we were able to reveal the circular structure of the nuclear pore (Fig. 3c–e). In addition, PA-SiR-Actin was tested for SMLM in fixed COS-7 cells revealing stress fibers and connecting thinner fibers (Supplementary Fig. 14c). Both PA-SiR-Halo and

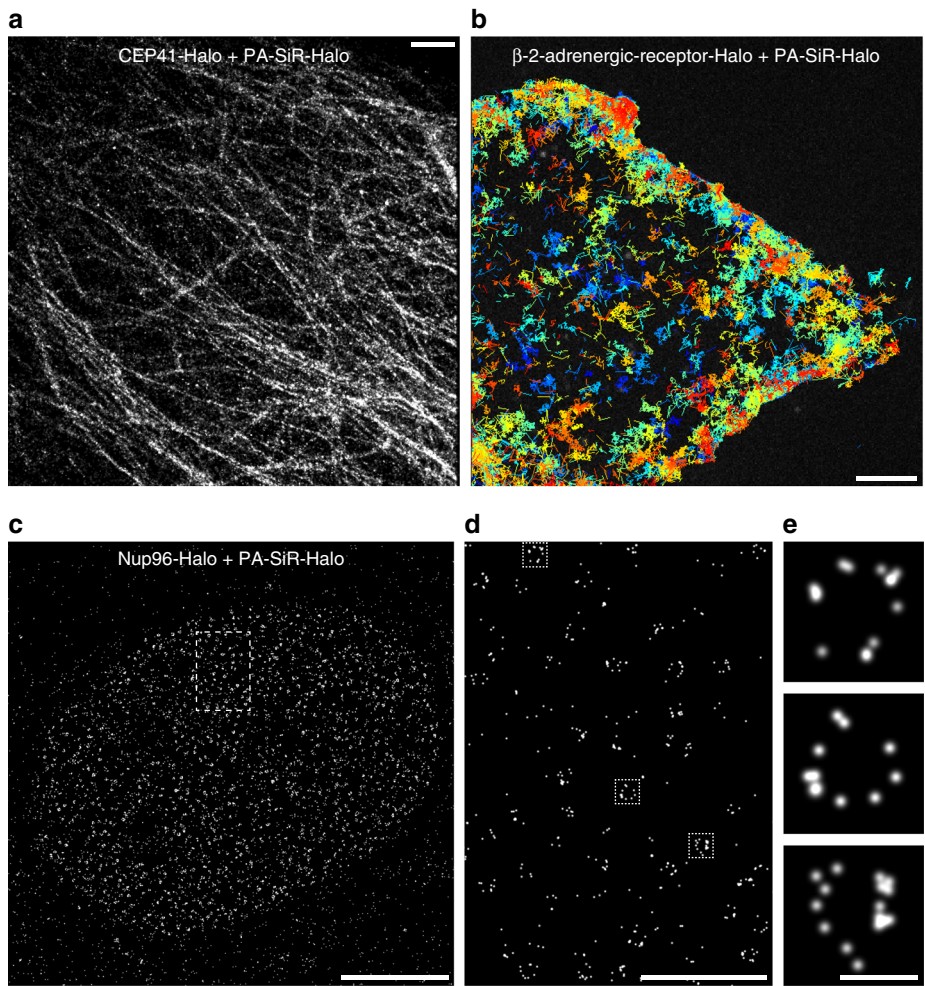

**Fig. 3** Super-resolution microscopy and single-particle tracking experiments. **a** Super-resolved image of microtubules in fixed U-2 OS cells stably expressing Cep41-Halo stained with PA-SiR-Halo (1 μM for 2 h). The image is reconstructed from 14,083 frames (100 ms exposure time, 2.9 kW cm$^{-2}$ at 642 nm excitation). The average microtubule diameter was found to be 38.7 ± 7.7 nm (mean ± 95% confidence interval, $N = 20$ tubules). Scale bar, 1 μM. **b** Image of cumulative single-particle tracks of β-2-adrenergic-receptor-Halo stained with PA-SiR-Halo (0.5 μM, 1 h) measured during 2 min. Continuous lines are drawn representing the movement of individual receptors. They are color coded in order to distinguish the individual tracks. For visibility, only tracks that have an overall displacement larger than 0.28 μm are shown (30 ms exposure time, 0.3 kW cm$^{-2}$ at 642 nm excitation). Scale bar, 5 μm. **c** Super-resolved overview image of the nuclear pore complex. Endogenously tagged Nup96-Halo in U-2 OS cells was stained with PA-SiR-Halo (1 μM for 2 h). Scale bar, 5 μm. **d** Super-resolved image from the boxed region in (**c**). Scale bar, 1 μm. **e** Single nuclear pores from boxed regions in (**d**) following the same order. Scale bar, 100 nm

PA-SiR-Actin are cell-permeable and make it possible to label live-cells, circumventing permeabilization steps during fixation and therefore reducing potential sources of artifacts[35].

**Live-cell single-particle tracking and SMLM**. We next tested the performance of PA-SiR-Halo in live-cell single-particle tracking photoactivated localization microscopy (sptPALM) (Fig. 3b)[36]. To this end, we chose to track a G-protein coupled receptor involved in cellular signaling that is located in the plasma membrane: beta-2-adrenergic receptor (β2AR)[37]. Using β2AR fused to HaloTag and labeled with PA-SiR-Halo, we were able to track β2AR for several hundreds of milliseconds before photobleaching (Fig. 3b). These track-lengths are considerably longer than what is commonly found for photoactivatable or photoconvertible proteins[20] and similar to what we found for PA-JF$_{646}$-Halo. Furthermore, β2AR labeled with either PA-SiR-Halo or PA-JF$_{646}$-Halo moved with comparable mean speeds (Supplementary Fig. 15). PA-SiR-Halo might prove to be beneficial over PA-JF$_{646}$-Halo in intracellular single-particle tracking experiments, where

high signal-to-background ratios are required. It has to be noted that the use of (high-intensity) UV light for photoactivation can cause fluorophore degradation and phototoxicity. UV light shows higher phototoxicity than red light[38,39] and should be used only at low intensities and/or low pulse frequencies. Phototoxicity caused by activation of small-molecule synthetic probes with UV light can be due to the UV light itself as well as toxic side products of the photoactivation. While PA-SiR does not release any toxic side products, the risk of conventional phototoxicity remains.

Finally, we investigated the potential of PA-SiR-Halo for live-cell SMLM. In such experiments, we could follow the fast dynamics of mitochondria (TOMM20-Halo) labeled with PA-SiR-Halo over one minute in 10 s snapshots without artificial narrowing and collapsing of structures (Fig. 4 and Supplementary Movie 1, Supplementary Fig. 16). The movie and the snapshots taken thereof revealed intermediate formation of thin tubules between mitochondria (blue arrowheads), as was previously seen with SMLM imaging of MitoTracker Red[40]. It was possible to follow fission events of mitochondria highlighting the dynamic

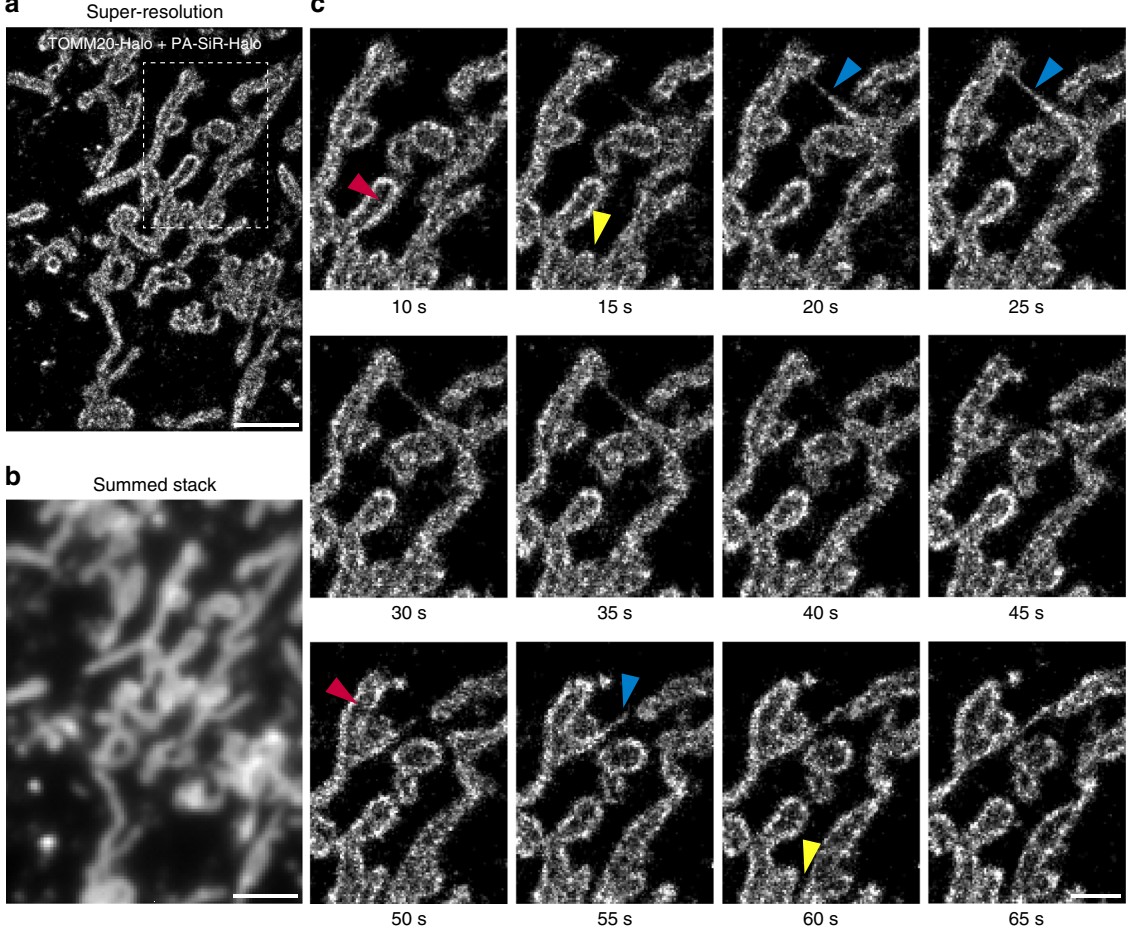

**Fig. 4** Live-cell SMLM of the outer mitochondrial membrane (TOMM20-Halo) labeled with PA-SiR-Halo. **a** Super-resolved image acquired within 10 s (50 ms exposure time, 0.3 kW cm$^{-2}$ 642 nm excitation). The recently published ImageJ plugin HAWK[71] was used to achieve imaging at high emitter densities to capture fast structural changes. **b** Sum projection over the first 10 s mimicking the diffraction limited image. Scale bar, 2 μm. **c** Time series of boxed region in (**a**). Each frame is reconstructed from 200 frames (10 s). For clarity, snapshots are shown only every 5 s. Several mitochondria are perceived to be hollow as Tomm20 is localized to the outer membrane of mitochondria (red arrowheads). The highly dynamic mitochondria form thin tubules between neighboring mitochondria (blue arrowheads) and disconnect (fission) in other areas (yellow arrowheads). Scale bar, 1 μm. Full rolling frame movie available as Supplementary Movie 1

network of connecting and disconnecting mitochondrial units (yellow arrowheads). Most interestingly, localizing the fluorophore to the outer membrane of the mitochondria further enabled us to distinguish the outer membrane from the matrix in several cases (red arrowheads), which has not been observed with live-cell SMLM so far. This will eventually help to study interactions between the inner and outer membrane of mitochondria by two color SMLM. This demonstrates that PA-SiR-Halo enables live-cell SMLM of intracellular targets.

## Discussion

In summary, PA-SiR is a photoactivatable, cell-permeable, far-red fluorophore that is activated by an unusual light-induced protonation. Its outstanding spectroscopic properties make it well suited for SMLM in both fixed and live-cells and enabled us to create powerful probes for HaloTag and actin. We expect that the exceptional properties of PA-SiR will be exploited in the future to create various other photoactivatable probes for live-cell imaging.

## Methods

**Synthesis and general methods**. Detailed procedures for the synthesis of all compounds and their characterization are given in the Supplementary Methods.

**General considerations PA-SiRs**. These were prepared as stock solutions in dry DMSO and diluted in the respective buffer such that the final concentration of DMSO did not exceed 5% v/v. A fiber coupled LED (Omicron, 340 nm, 3 mm liquid light guide) was used to perform UV irradiation unless otherwise stated. Phosphate-buffered saline (PBS) (6.7 mM, Lonza) was used in all experiments.

**UV-Vis measurements**. All absorbance measurements (spectra and time traces) were performed in 1.5 mL stirrable quartz cuvettes (Hellma Analytics) on a JASCO V770 spectrophotometer with a Peltier element (PAC743R) under continuous stirring and at 21 °C. PA-SiRs were diluted in PBS (10 μM unless otherwise stated). A blank was measured before starting the measurement. UV irradiation was performed directly inside the spectrophotometer during the ongoing experiment for 12 s unless otherwise stated. PBS solutions of different pH were adjusted by addition of HCl or NaOH solution using a pH meter. Cysteamine concentrations were adjusted by the addition of concentrated cysteamine solution (1 M). PA-SiR-Halo, PA-SiR-SNAP and PA-SiR-Actin probes (10 μM) were directly added to the target protein (20 μM SNAP-tag, 20 μM HaloTag or 0.4 mg mL$^{-1}$ G-actin), or to a bovine serum albumin (Sigma) solution in PBS. The mixture was incubated for 1 h (HaloTag) or 2 h (SNAP-tag) at room temperature. In the case of the actin probe, buffer containing 5 mM Tris-HCl (pH 8.0), 0.2 mM CaCl$_2$ and 0.2 mM ATP was used. This buffer was supplemented with 50 mM KCl, 2 mM MgCl$_2$, 5 mM guanidine carbonate and 1 mM ATP to obtain F-actin. Both buffers are components of the actin polymerization fluorescence assay kit (Cytoskeleton). The samples were incubated for 2–3 h at 37 °C. Measurements were performed in triplicates expect for the saturation experiments with 405 nm irradiation these were performed in duplicates. Fitted parameters such as decay constants etc. are reported as the average of three fits. Representative measurements are displayed. Where given $X^2$ the reduced chi-squared corresponds to the residual sum of square (RSS) and $R^2$ is

the squared correlation coefficient. They are defined as follows:

$$X^2 = \text{RSS} = \sum_{i=1}^{n} (y_i - \hat{y}_i)^2, \tag{1}$$

$$R^2 = 1 - \frac{\text{RSS}}{\text{TSS}} = 1 - \frac{\sum_{i=1}^{n} (y_i - \hat{y}_i)^2}{\sum_{i=1}^{n} (y_i - \overline{y}_i)^2}. \tag{2}$$

**Fluorescence spectra and quantum yield.** Fluorescence spectra were measured on a JASCO FP-8600 fluorimeter in 1.4 mL fluorescence cuvettes (Hellma Analytics). Emission spectra were collected from 610 to 1000 nm exciting at 580 nm; excitation spectra were recorded at 664 nm exciting from 400 to 655 nm unless otherwise stated. Fluorescence intensity upon addition of cysteamine was measured on a plate reader (TECAN Spark® 20 M) equipped with a monochromator exciting at 640/10 nm and collecting the emission at 670/10 nm. Quantum yields were determined using a Hamamatsu Quantaurus QY.

**Extinction coefficient and quantum yield of activation.** Extinction coefficients at 646 nm after activation were calculated from the equilibrium constants ($K_2$) obtained in the 12 s activation experiments (Fig. 2b, Supplementary Figs. 6 and 10, Table 5 and 7), assuming that during the activation the decay ($k_2$ and $k_{-2}$) is negligible, and the absorbance reached its equilibrium in the saturation experiment (Supplementary Fig. 11, Table 8). Quantum yields of activation were determined using standard ferroxalate actinometry[41] along with the activation rates determined in the saturation experiments (Supplementary Fig. 11). This calculation does not take into account the decay kinetics but was good enough to give an estimate of the quantum yields of activation. Saturation experiments under 405 nm irradiation (Supplementary Fig. 11e, f) were performed using a Solis405C High-Power LED (Thorlabs) coupled to a 3 mm liquid light guide.

**Computational chemistry.** Optimization of the PA-SiR structure as well as HOMO/LUMO calculations were performed at the B3LYP/6-31G(d) level of theory by using the software package Gaussian 09[42].

**X-ray crystal structure determination.** Colorless needle shaped crystals were grown from slow evaporation of a $CH_2Cl_2$/MeOH solution at 4 °C.

Crystal data and details of the structure determinations are compiled in Supplementary Table 11. A full shell of intensity data were collected at low temperature with an Agilent Technologies Supernova-E CCD diffractometer (Mo-$K_\alpha$ radiation, microfocus X-ray tube, multilayer mirror optics). Detector frames (typically ω- occasionally φ-scans, scan width 0.5°) were integrated by profile fitting[43,44]. Data were corrected for air and detector absorption, Lorentz and polarization effects[43] and scaled essentially by application of appropriate spherical harmonic functions[43,45,46]. Absorption by the crystal was treated numerically (Gaussian grid)[45,47]. An illumination correction was performed as part of the numerical absorption correction[45]. The structures were solved by ab initio dual space methods involving difference Fourier syntheses (VLD procedure)[48,49] and refined by full-matrix least squares methods based on $F^2$ against all unique reflections[50–53]. All nonhydrogen atoms were given anisotropic displacement parameters. The positions of most hydrogen atoms (except those of the methyl groups, which were treated as variable metric rigid groups with local $C_3$ symmetry) were taken from difference Fourier syntheses and refined.

CCDC 1942173 contains the supplementary crystallographic data for this paper. These data can be obtained free of charge from the Cambridge Crystallographic Data Centre's and FIZ Karlsruhe's joint Access Service via https://www.ccdc.cam.ac.uk/structures/?

Visualization was performed using ORTEP III[54] and POV-Ray 3.7.0[55].

**LC-MS analysis.** Liquid chromatography mass spectrometry (LC-MS) was performed on a Shimadzu MS2020 connected to a Nexera UHPLC system equipped with a Supelco Titan C18 80 Å (1.9 µm, 2.1 × 50 mm) column. Buffer A: 0.05% HCOOH in $H_2O$ Buffer B: 0.05% HCOOH in ACN. PA-SiR was dissolved in MQ water (~20 µM). UV irradiation was performed for 1 min in a quartz cuvette (Hellma Analytics) and aliquots were taken to measure LC–MS at defined time points using an analytical gradient from 10 to 90% B within 6 min with 0.5 mL min$^{-1}$ flow.

**¹H NMR analysis PA-SiR.** PA-SiR (1 mg, 2.0 µmol) was dissolved in PBS/$D_2O$ (1 mL, 90:10) and NaOH (1 µL, 5 M) was added to achieve better solubility as PA-SiR was isolated as its TFA salt (pH = 7–8, pH paper). ¹H NMR spectra were measured on a Bruker AV 600 spectrometer at 600 MHz and 298 K. Chemical shifts δ are reported in ppm downfield from tetramethylsilane using the DMSO signal ($\delta_H$ = 2.50 ppm) instead of the residual deuterated solvent signal as an internal reference. Spectra were measured with NS = 128 using a water suppression presaturation sequence. UV irradiation was performed outside of the spectrometer for the indicated times with a transilluminator (Biometra TI 1, 312 nm). After each UV irradiation step the NMR sample was transferred to the NMR spectrometer.

**Plasmids.** A pcDNA5/FRT/TO vector (ThermoFisher Scientific) was used for transient expression in mammalian cells and generation of stable cell lines. A pET51b(+) vector (Novagen) was used for protein production in *Escherichia coli*. Proteins were tagged Strep and His$_{x10}$ N- and C-terminal, respectively. SNAP-tag and HaloTag7 were fused to the N or C terminus of the genes of interest (GOI) and a T2A-EGFP sequence was introduced. Cloning was performed by Gibson assembly[56]. GOI: H2B (NEB, pSNAPf-H2B), CEP41 (Genecopoeia (GC-V1653 and GC-V1653-CF))[22], mEOS3.2 (Addgene #54525)[57], Lifeact (Addgene #36201)[58], TOMM20 (Addgene #55146, gift from Michael Davidson), β-2-adrenergic-receptor-Halo (Addgene #66994, gift from Catherine Berlot) were used as entry plasmids.

**Protein production and purification.** Proteins were expressed in *Escherichia coli* strain BL21(DE3)-pLysS. Luria–Bertani broth cultures were grown at 37 °C to optical density at 600 nm (OD$_{600nm}$) of 0.8, induced by the addition of 0.5 mM isopropyl-β-D-thiogalactopyranoside and grown at 17 °C overnight in the presence of 1 mM $MgCl_2$. The cells were harvested by centrifugation (4500 g, 10 min, 4 °C) and lysed by sonication. The cell lysate was cleared by centrifugation (20,000 g, 20 min, 4 °C). All proteins were purified using affinity-tag Ni-NTA (Qiagen) leading to higher than 95% pure proteins (verified by sodium dodecyl sulfate polyacrylamide gel electrophoresis (PAGE) coomassie staining). mEos3.2-Halo was purified analogously but using an additional Strep-Tactin (IBA) column purification step to reach higher purity and following the suppliers' instructions. Proteins were finally concentrated using an Ultra-0.5 mL centrifugal filter device (Amicon) with a molecular weight cut-off according to the protein size and then stored in a glycerol 45% (v/v) solution at −20 °C. Proteins were used from glycerol stocks and were further diluted. The amino acid sequences can be found in the Supplementary Methods.

**Polyacrylamide gel electrophoresis.** HaloTag protein (4 µM) was labeled using PA-SiR-Halo (0, 1, 2, 3, 4, or 6 µM) in activity buffer (50 mM HEPES, 50 mM NaCl, pH 7.3) for 2 h at room temperature. After labeling, the proteins were separated by PAGE (4–20% 10-well Mini-Protean TGX, BioRad) as recommended by the manufacturer and revealed by in gel fluorescence using a ChemiDoc MD Imaging System (BioRad). PA-SiR-Halo labeled proteins were imaged using red epi illumination (695/55 nm). PA-SiR-Halo was activated using the UV-transilluminator of the ChemiDoc MD Imaging System.

**Cell culture and transfection.** HeLa, U-2 OS (both ATCC), COS-7 (Gift from Dr. R. Sprengel, MPI for Medical Research) or U-2 OS NUP96-Halo (generously provided by the Ellenberg lab, EMBL) cells were cultured in high-glucose phenol-red free Dulbecco's Modified Eagle Medium (DMEM) (Life Technologies) medium supplemented with GlutaMAX (Life Technologies), sodium pyruvate (Life Technologies) and 10% fetal bovine serum (FBS) (Life Technologies) in a humidified 5% $CO_2$ incubator at 37 °C. Cells were split every 3–4 days or at confluency. These cell lines were regularly tested for mycoplasma contamination. Cells were seeded on glass bottom 35 mm dishes (Mattek or Greiner bio-one), 10-well glass bottom dishes (Greiner bio-one) or 24 mm high precision round coverslips #1.5 (Carl Roth GmbH) one day before imaging. Transient transfection of cells was performed using Lipofectamine™ 2000 reagent (Life Technologies) according to the manufacturer's recommendations: DNA (2.5 µg) was mixed with OptiMEM I (100 µL, Life Technologies) and Lipofectamine™ 2000 (6 µL) was mixed with OptiMEM I (100 µL). The solutions were incubated for 5 min at room temperature, then mixed and incubated for additional 20 min at room temperature. The prepared DNA–Lipofectamine complex was added to a glass bottom 35 mm dish with cells at 50–70% confluency. After 12 h incubation in a humidified 5% $CO_2$ incubator at 37 °C the medium was changed to fresh medium. The cells were incubated for 24–48 h before imaging.

**Stable cell line establishment.** The Flp-In™ T-REx™ System (ThermoFisher Scientific) was used to generate stable cell lines exhibiting tetracycline-inducible expression of the gene of interest (GOI). Briefly, pcDNA5-FRT-TO-GOI and pOG44 were co-transfected into the host cell line U-2 OS FlpIn TREx[59]. Homologous recombination between the FRT sites in pcDNA5-FRT-TO-GOI and on the host cell chromosome, catalyzed by the Flp recombinase expressed from pOG44, produced the U-2 OS FlpIn TREx cells expressing stable and inducible the GOI. Selection was performed using 100 µg mL$^{-1}$ hygromycin B (ThermoFisher Scientific) and 15 µg mL$^{-1}$ blasticidine (ThermoFisher Scientific). Stable cell lines were seeded on glass bottom dishes as described in the previous section, and induced using 100 µg mL$^{-1}$ doxycycline (Sigma-Aldrich) for 24–48 h previous to imaging.

**Staining.** Cells were stained with 0.2–1 µM PA-SiR (1–2 h, 37 °C) in phenol-red free DMEM medium supplemented with GlutaMAX, sodium pyruvate and 10% FBS (all Life Technologies), washed with the same medium or PBS (once for 3 min, 37 °C) and imaged in the same medium.

**Sample preparation for super-resolution microscopy.** U-2 OS-CEP41-Halo cells were seeded on 24 mm glass coverslips and stained with PA-SiR as described above.

Methanol fixation was performed as follows: growth medium was removed, cells were incubated for 7 min in −20 °C cold methanol and washed twice with PBS. Fixed-cell samples were mounted in PBS on cavity slides (VWR™) sealed with twinsil® 22 (Picodent) and imaged therein.

COS-7 cells were seeded on 24 mm glass coverslips and stained with PA-SiR-Actin as described above. The cells were fixed as previously described[60]. Briefly, they were fixed and extracted for 1 min using a solution of 0.3% [w/v] glutaraldehyde and 0.25% [v/v] Triton X-100 in CB buffer (CB: 10 mM MES, pH 6.1, 150 mM NaCl, 5 mM EGTA, 5 mM glucose and 5 mM $MgCl_2$), and then postfixed for 10 min in 2% [w/v] glutaraldehyde in CB. They were treated with freshly prepared 0.1% sodium borohydride for 7 min. Short additional poststaining was performed with 0.5 μM PA-SiR-Actin (1 h, 25 °C). Fixed-cell samples were mounted in PBS on cavity slides (VWR™) sealed with twinsil® 22 (Picodent) and imaged therein.

U-2 OS cells were seeded on 24 mm glass coverslips and transiently transfected (TOMM20-Halo or β-2-adrenergic-receptor-Halo). The next day the cells were stained with PA-SiR-Halo as described above and the coverslips were mounted into attofluor cell chambers (Life technologies) and the imaging medium was supplemented with HEPES (20 mM). Cells were directly imaged after mounting.

Genome-edited U-2 OS cells with Halo-tagged NUP96[32] were seeded on 24 mm round coverslips (No. 1.5 H; 117640; Marienfeld). Cells were cultured under adherent conditions at 37 °C, 5% $CO_2$ and 100% humidity in DMEM (high glucose, without phenol red) supplemented with 10% [v/v] FBS, 2 mM L-glutamine, nonessential amino acids, and ZellShield. Before sample preparation, the respective dye was added to the medium to a final concentration of 1 μM and incubated for 2 h. All following incubations were carried out at room temperature and all incubations longer than 1 min were performed on an orbital shaker in the dark to prevent preactivation of the dye. Cells were prefixed in 2.4% [w/v] formaldehyde (FA) in PBS for 30 s, permeabilized in 0.4% [v/v] Triton X-100 in PBS for 3 min and fixed in 2.4% [w/v] FA in PBS for 30 min. Subsequently, the FA was quenched by incubating the coverslip for 5 min in 100 mM $NH_4Cl$ in PBS. After washing three times for 5 min each in PBS, the coverslips were mounted and imaged in PBS.

**Single-molecule assay.** Sample preparation was adapted from two literature procedures[61,62]. Briefly, 18 × 18 mm high-precision coverslips (Carl Roth) were sonicated for 10 min in MQ water, 10 min in acetone, 10 min in MeOH, 10 min in KOH (1 M, prepared from 99.98% purity Carl Roth) and rinsed with MQ water after each step. The coverslips were cleaned with piranha solution (1:3, $H_2O_2$/$H_2SO_4$) twice for 30 min. After extensive rinsing with MQ water they were dried under a $N_2$ stream. A solution of 2% [v/v] N-[3-(trimethoxysilyl)propyl]ethylendiamine (Sigma-Aldrich) in dry acetone was prepared and the clean coverslips were immersed in the dark for 1 h. The coverslips were rinsed with acetone, MQ water and then dried with $N_2$. A solution of 1 mg biotin-PEG-SVA (MW 5000, Laysan Bio) and 54 mg mPEG-SVA (MW 5000, Laysan Bio) was prepared in 230 μL sodium bicarbonate buffer (10 mM freshly prepared) and applied to three coverslip pairs. After 3 h in the dark the coverslips were washed with MQ water, blow dried with $N_2$ and stored under $N_2$ at −20 °C. Flow chambers were assembled at need from one glass slide (Carl Roth) and one coated coverslip separated by double sided tape and fixed with epoxy glue.

In total, 100 μL of a 0.2 mg mL$^{-1}$ solution of streptavidin (Life Technologies) in PBS was applied to the flow chamber and incubated for 10 min. The channel was washed with 400 μL PBS. A solution of SNAP-tag:EGFP:HaloTag (5 μM), fluorophore (2.5 μM), biotin-ligand (5 μM; SNAP-Biotin™ (NEB), HaloTag Biotin (Promega)), in PBS, was prepared and incubated for 1 h. In total, 100 μL of a 1:1000–1:500 dilution thereof was applied to the flow chamber and incubated for 10 min. The channel was washed with 400 μL PBS and filled with PBS. They were imaged in TIRF mode using a Leica SR GSD (Supplementary Table 10).

**Widefield microscopy.** Imaging was performed using a Leica DMi8 microscope (Leica Microsystems) equipped with a Leica DFC9000 GT sCMOS camera; a CoolLED Pe4000 LED light source (635 nm, 635/18; 470 nm, 474/27; 365 nm, 378/52); a HC PL APO 40.0 ×/1.10 water objective and standard GFP (515/40) and Cy5 (720/100) filter sets. Activation of the fluorophores was achieved by irradiation with the 365 nm LED and the DAPI filter set (430/35) at 100% LED output for the indicated durations. The microscope was equipped with a $CO_2$ and temperature controllable incubator (PeCon, 37 °C).

For stability measurement images were taken in the Cy5 (500 ms, ex: 10%), transmission (100 ms) and the GFP channel (100 ms, ex: 5%) every 30 s. Activation was performed for 1 s once.

For activation experiment images were taken in the Cy5 (500 ms, ex: 10%), transmission (100 ms) and the GFP channel (100 ms, ex: 5%) consecutively every 9 s. Activation was performed for 50 ms after each acquisition cycle.

**Confocal microscopy.** Confocal imaging was performed on a Leica DMi8 microscope (Leica Microsystems) equipped with a Leica TCS SP8 X scanhead; a SuperK white light laser, a 355 nm CW laser (Coherent), a HC PL APO 63 ×/1.47 oil objective or a HC PL APO 40.0 ×/1.10 water objective; emission was collected as indicated in Supplementary Table 10. Photoactivation was performed for one

frame by using a 355 nm laser. The microscope was equipped with a $CO_2$ and temperature controllable incubator (Life Imaging Services, 37 °C).

For signal to background measurement cells were focused in the transmission channel and z-stacks were recorded with 0.4 μm step size before and after activation. The summed stacks were analyzed as follows: the mean of a rectangular ROI within the nucleus was divided by the mean of a rectangular ROI adjacent to the nucleus.

**Super-resolution microscopy.** Super-resolution microscopy was performed on a Leica SR GSD (Leica Microsystems) microscope equipped with an Andor iXon3 897 EMCCD camera (Andor) using a central 180 × 180 pixel or 400 × 400 pixel subregion of the camera chip. The system was equipped with the following lasers for excitation and photoactivation: a 642 nm (500 mW; MPBC, Inc.), a 532 nm (1000 mW; MPBC, Inc.), a 488 nm (500 mW; MPBC, Inc.), and a 405 nm (30 mW; Coherent, Inc.) diode laser for photoactivation. The standard Leica filter sets for SR GSD systems were used—in brief: Leica set 488 for 405 and 488 nm excitation: DBP 405/10 488/10 excitation filter, LP 505 dichroic mirror and 555/100 suppression/emission filter; Leica set 532 for 405 and 532 nm excitation: DBP 405/10 532/10 excitation filter, LP 550 dichroic mirror and 600/100 suppression/emission filter; Leica set 642 for 405 and 642 nm excitation: DBP 405/10 642/10 excitation filter, LP 650 dichroic mirror and 710/100 suppression/emission filter. Fluorescence was collected through a high-numerical-aperture (NA) oil-immersion objective (Leica HC PL APO 160 ×/1.43). Lateral drift was minimized by the suppressed motion (SuMo) stage of the Leica SR GSD and by keeping the temperature of the environment stable via an incubation box ($T = 21 \pm 0.1$ °C, instrument parameter) covering the entire microscope. The microscope was operated by the Leica LAS X software (version 1.9.0.13747). The specific parameters can be found in Supplementary Table 10.

NUP96-Halo samples were imaged on a custom-built epi-fluorescence microscope with homogenous high-power illumination[63]. The output of a commercial LightHub laser box (Omicron-Laserage Laserprodukte) with 405, 488, 561, and 640 nm laser lines and an additional 640 nm booster laser (Toptica) were focused on a speckle reducer (LSR-3005-17S-VIS; Optotune) and coupled into a multi-mode fiber (M105L02S-A; Thorlabs). The output of this fiber is magnified by an achromatic lens, cleaned up by a quadband filter (390/482/563/640 HC Quad; AHF) and focused into the sample. Fluorescence was collected through a high-numerical aperture (NA) oil-immersion objective (160×/1.43 NA; Leica), filtered by a 700/100 bandpass filter (AHF) and focused onto an Evolve512D EMCCD camera (Photometrics). The focus was stabilized by a total internally reflected IR laser that was focused onto a quadrant photodiode, which was coupled into a closed-loop with the piezo objective positioner. Typically, we acquire 15,000–30,000 frames with 50 ms exposure time and laser power densities of about 13 kW cm$^{-2}$. Data were acquired until no more activated fluorophores were observed. The pulse-length of the 405 nm laser was adjusted during the acquisition to maintain a similar number of localizations per frame. The different components of the microscope are managed by a field-programmable gate array (Mojo; Embedded Micro) which is controlled using a custom-written plugin for μManager[64].

**Software and image processing.** Statistical analysis as well as curve fitting was performed using OriginLab[65]. All images except the NUP96-Halo images were processed with ImageJ/Fiji[66,67]. Super-resolution images and TIRF data from the single-molecule assay were processed with the ImageJ plugin ThunderSTORM[68]. Tracking data were analyzed using the TrackMate plugin[69]. Single-molecule assay data were further processed by a costume written MatLab script provided by Dr. Christian Sieben (EPFL) based on the Crocker, Weeks, and Grier Algorithm[70]. For the quantification of microtubule width, we constructed a perpendicular line profile from a 250 nm long section of the microtubule. We then fitted a Gaussian distribution (bin width 2 nm) to the profile and plotted its FWHM in a boxplot (Supplementary Fig. 14g, $N = 20$ line segments per dye).

$$y(x) = y_0 + A \cdot e^{-(x-x_c)^2/2 \cdot s^2}, \qquad (3)$$

$$\text{FWHM} = 2 \cdot \sqrt{2 \cdot \ln 2} \cdot s. \qquad (4)$$

Live-cell SMLM data were additionally processes using the HAWK plugin using three levels and time grouping, followed by multi-emitter fit in ThunderSTORM allowing for five emitters per fitting region[71]. Costume written MatLab code was used to produce the rolling frame video. The movie presented was convoluted with a Gaussian function (sigma = 12 nm).

The reconstruction of super-resolved images of NUP96-Halo was done using the custom-written software SMAP (Super-resolution Microscopy Analysis Platform, https://github.com/jries/SMAP). First, localizations were detected using a difference of Gaussians algorithm and a dynamic threshold to exclude random signal fluctuations. Then the localizations were fit by a pixelated Gaussian function. Dim localizations (localization precision > 30 nm) and out-of-focus localizations (fitted size of the Gaussian > 160 nm) were filtered out. Localizations that were found within 75 nm of each other in consecutive frames with maximum one frame dark time were grouped into one localization. Images were reconstructed by plotting all localized emitters at the fitted positions as Gaussians with a width proportional to their localization precision.

**Reporting summary**. Further information on research design is available in the Nature Research Reporting Summary linked to this article.

## Data availability

The data supporting the findings of this study are available within the paper and its Supplementary Information and are available from the corresponding author upon reasonable request. The source data underlying Figs. 1e and 2c, Supplementary Figs. 9, 10c, 12d–f, 13b, 14g, and 15a–c and Supplementary Tables 1, 2, 5, 7, and 9 are provided as a Source Data file.

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

## Acknowledgements
This work was supported by the Max Planck Society, the École Polytechnique Fédérale de Lausanne, a grant from the Swiss Commission for Technology and Innovation (CTI), the NCCR Chemical Biology, and the European Molecular Biology Laboratory (to P.H., M.L., B.N., M.K., J.E. and J.R.), the EMBL International PhD Program (to P.H.), the European Research Council (ERC CoG-724489, to P.H. and J.R.), and the National Institutes of Health Common Fund 4D Nucleome Program (Grant U01 EB021223/U01 DA047728 to J.E. and J.R.). All requests for the NUP96-Halo cell line should be directed to Jan Ellenberg. The authors thank Dr. H. Farrants, Dr. J. Hiblot for sharing reagents, Dr. B. Koch for help with the establishment of the stable CEP41-Halo cell line, Dr. C. Sieben (EPFL) for valuable discussions and sharing of the Matlab analysis script, Dr. Rolf Sprengel (MPI for Medical Research) for the donation of the COS-7 cells, the electronic workshop of the Max Planck Institute for Medical Research for technical assistance, the NMR service of EPFL for assistance with the NMR experiments, Heidrun Haungs for technical assistance with the X-ray crystallography, and the Advanced Light Microscopy Facility (ALMF) at the European Molecular Biology Laboratory (EMBL) and Leica Microsystems for support.

## Author contributions
M.S.F., P.H., M.L., J.R., S.P., L.R. and K.J. planned the experiments and co-wrote the paper. L.R. made the first observation of PA-SiR photoconversion and originated the project. M.S.F. performed the chemical synthesis and characterization as well as the widefield and confocal measurements. M.S.F. performed the SMLM on CEP41-Halo, F-actin, and mitochondria with assistance from M.L. B.N., M.K. and J.E. provided the U-2 OS NUP96-Halo cell line. M.S.F. and P.H. performed the SMLM on NUP96-Halo. H.W. solved the crystal structure.

## Competing interests
M.S.F., S.P., L.R. and K.J. are inventors on a patent filed by EPFL and Spirochrome AG. Remaining authors declare no competing interests.
