## [Peer Review File · Nature Communications]

Reviewers' comments:

Reviewer #1 (Remarks to the Author):

The paper describes the properties of PA-SiR, a new variant of silicon rhodamine. Labelling is one of the major challenges in SMLM, in particular when it comes to live cell imaging, where phototoxicity is a major issue. SiR has been used for STED and there is some evidence suggesting that it has a lower phototoxic effect than organic dyes, but SiR is not suitable for SMLM because it hardly blinks. The authors have managed to generate a version of SiR that can be photoactivated, which is more useful than blinking (which is often used for SMLM) because it is much more controllable. This is an important development which I have no doubt will be useful to many labs. There is considerable evaluation of the properties of the probe in the paper, which clearly demonstrate how it will perform in a variety of situations. It is particularly impressive that they are able to obtain good super-resolution images in live cells. The results are absolutely deserving of publication.

I have some minor criticisms which I would like to see addressed:

- 1) I think it is worth expanding the first part of the introduction a little. At the moment it just states that 'small molecule fluorophores are generally brighter and more stable than fluorescent proteins', but I think at this point it would be worth discussing why they are still used, even for SMLM: ease of use, the challenge of getting dyes across the membrane, and the excellent controllability of (some of the) photoswitchable proteins. This will help to make it more obvious why the results are a major advance.
- 2) With regard to the Janelia fluorophores, what is the 'dark side product'? Does it have any effect on the cell?
- 3) In figures the label is given (e.g. CEP41, TOMM20) but I think the thing that is being labelled (microtubules, membrane of the mitochondria) should be given in the first sentence of the figure captions, again for clarity.
- 4) Why is the quantum yield given for 340nm when in practice it will almost certainly be used with 405nm? The paper cited here also seems to use 405nm. The table cited here also doesn't show that the values are similar to fluorescent proteins, I think it could be useful to include some other fluorophores in the table to enable easy comparison.
- 5) I am not keen on microtubule diameter as a measure, but if give I think it's worth giving the context of how big the microtubule itself is expected to be and how big the labelling system is expected to be.
- 6) In supplementary figure 9 the transmission images are almost impossible to distinguish any features – perhaps the minimum/maximum displayed values can be adjusted with some type of normalisation? At the moment the images do not convey information.
- 7) What is the impact of the UV needed for photoactivation vs. by-products of the dark state when an organic dye is used in terms of phototoxicity? There must be some sort of tradeoff, and at the moment that isn't clear. I don't want to suggest further experiments as I think there is sufficient evaluation and phototoxicity assessments are very challenging, but I think it should be discussed.

Reviewer #2 (Remarks to the Author):

In this work, the author developed a photoactivatable silicon rhodamines featured with π -conjugation converting, which is employed for live-cell super-resolution microscopy. However, the mechanism of a light-dependent isomerization followed by protonation for spectroscopic properties is hard to be proven with limited the structural information of PA-SiR. In order to address a broader audience, solid evidence of structure and clear illustration of mechanism must be presented to clarify that.

Comments:

1. Light-dependent isomerization followed by protonation is critical for this work. However, the only calculated result is not the compelling evidence to prove the structure of "PA-SiR 4" and the isomerization of PA-SiR. The authors should provide solid and clear evidences, including single crystal X-ray diffraction etc., to illustrate the conversion of π -conjugation.
2. In page 4, the authors mentioned that "PA-SiR-Halo attached to HaloTag can be very efficiently activated". But the mechanism is unclear and confusing. More detailed discussion need about the

reason why PA-SiR-Halo attached to HaloTag is more stable than the isolated one.

3. The authors mentioned that "we prepared conjugates of PA-SiR with ligands for protein labeling". There are a few discussions about conjugation between PA-SiR-Halo and protein. Please provide more details like gel electrophoresis to prove that PA-SiR-Halo has conjugated with HaloTag.

Reviewer #3 (Remarks to the Author):

"Photoactivation of silicon rhodamines via a light-induced protonation" by Reymond, Johnsson and co-workers describes a photoinduced mechanism to activate a cell-permeable, far-red fluorophore ideal for super-resolution microscopy. From a serendipitous discovery in an attempted synthesis of a silicon rhodamine with an alkyl chain, a protonated non-fluorescent form was isolated. Interestingly, UV irradiation induced the formation of the fluorescent form having outstanding spectroscopic properties. The behavior of the fluorophore was subsequently investigated in great detail and halo-, actin and SNAP derivatives were prepared. Super-resolution microscopy, single-particle tracking experiments and live-cell microscopy revealed a clearly improved behavior, signal-to-background ratio, high permeability and stability.

Overall, the manuscript describes a highly valuable photoactivatable fluorophore with remarkable spectroscopic properties and distinctly enhanced utility in super-resolution microscopy. It is therefore recommended to accept this paper for publication after revisions taking in consideration the following suggestions and comments.

Main comments:

(1) The first analogue of this new class was found serendipitously discovered: the synthetic scheme of preparation would definitely improve clarity when illustrated to the main article.

(2) The yields for PA-SiR (SiRA in some SI schemes) and PA-SiR C3 is 5% and 8%, resp.: can these yields be further improved? An explanation of these results should be provided.

(3) What is the analogous behavior of related fluorophores such as fluorescein or rhodamine and was previously reported (e.g. J. Org. Chem. 2013, 78, 1833)?

(4) For the conceptual basis of the new class of fluorophores: what is the broader scope of the photoactivation strategy?

(5) An isomerization/protonation sequence is mentioned in the manuscript: the isomerization step is however not clear and an intramolecular charge-transfer is suggested based on calculations. Is an isomerization required or is protonation taking place directly upon charge-transfer? Otherwise a mechanism shown in Fig.1a would clearly improve the model.

Minor comment:

Some schemes in the SI use a different nomenclature for PA-SiRs (SiRA)

POINT-BY-POINT RESPONSE TO THE COMMENTS ON THE MANUSCRIPT “PHOTOACTIVATION OF SILICON RHODAMINES VIA A LIGHT-INDUCED PROTONATION”

We are grateful to the reviewers and editor for their insightful comments and suggestions. In the revised manuscript we have attempted to address all these points and changed the manuscript accordingly. Additionally, the manuscript was revised according to the *Nature Communications* style guide. To facilitate the reviewing of the revised manuscript we have provided a manuscript file with highlighted changes as well as a final clean version. Below we provide a point-by-point response to the comments of the reviewers.

Reviewer #1 (Remarks to the Author):

The paper describes the properties of PA-SiR, a new variant of silicon rhodamine. Labelling is one of the major challenges in SMLM, in particular when it comes to live cell imaging, where phototoxicity is a major issue. SiR has been used for STED and there is some evidence suggesting that it has a lower phototoxic effect than organic dyes, but SiR is not suitable for SMLM because it hardly blinks. The authors have managed to generate a version of SiR that can be photoactivated, which is more useful than blinking (which is often used for SMLM) because it is much more controllable. This is an important development which I have no doubt will be useful to many labs. There is considerable evaluation of the properties of the probe in the paper, which clearly demonstrate how it will perform in a variety of situations. It is particularly impressive that they are able to obtain good super-resolution images in live cells. The results are absolutely deserving of publication.

I have some minor criticisms which I would like to see addressed:

1) I think it is worth expanding the first part of the introduction a little. At the moment it just states that ‘small molecule fluorophores are generally brighter and more stable than fluorescent proteins’, but I think at this point it would be worth discussing why they are still used, even for SMLM: ease of use, the challenge of getting dyes across the membrane, and the excellent controllability of (some of the) photoswitchable proteins. This will help to make it more obvious why the results are a major advance.

Response: We thank reviewer #1 for the suggestion. We extended the introduction and commented on the use of fluorescent proteins (page 2).

2) With regard to the Janelia fluorophores, what is the ‘dark side product’? Does it have any effect on the cell?

Response: We specified the nature of the dark side-product in the main text (page 2, line 16) and depicted its structure in the new Supplementary Figure 1. To the best of our knowledge, the toxicity of the dark side-product itself was not yet investigated. However, the activation reaction of PA-JF proceeds through a reactive carbene intermediate, which has been shown to react with nearby nucleophiles (reference 21 of revised manuscript).

3) In figures the label is given (e.g. CEP41, TOMM20) but I think the thing that is being labelled (microtubules, membrane of the mitochondria) should be given in the first sentence of the figure captions, again for clarity.

Response: We implemented the reviewer's suggestion both in the main text figure captions (Fig.2, 3 and 4) and in the supplement (12 and 14).

4) Why is the quantum yield given for 340nm when in practice it will almost certainly be used with 405nm? The paper cited here also seems to use 405nm. The table cited here also doesn't show that the values are similar to fluorescent proteins, I think it could be useful to include some other fluorophores in the table to enable easy comparison.

Response: The quantum yield of photoactivation was measured with a light source at 340 nm as photoactivation at this wavelength is much more efficient. When we tried to measure this value with irradiation at 405 nm *in vitro*, extremely slow activation kinetics were observed (see new Supplementary Figure 11). This made a precise determination of the quantum yield of activation for PA-SiR and PA-SiR-C3 impossible as the decay kinetics are competing with activation. Nonetheless, we used these data to estimate the quantum yield of activation for PA-SiR-Halo on HaloTag and included these values in the main text (page 4, bottom). The corresponding data can be found in Supplementary Figure 11 and Tables 1, 2 and 9. For comparison, we also estimated the quantum yield of photoactivation for PA-JF₆₄₆ at 405 nm, which is comparable to that of PA-SiR-Halo. It should be noted that also for PA-JF₆₄₆ a precise determination of the quantum yield of activation was difficult as the decay kinetics (bleaching and chemical degradation) are competing with activation. We also took up the other suggestion of reviewer #1 and listed literature values for quantum yields of activation for relevant small-molecule fluorophores and fluorescent proteins in Supplementary Table 2 of the revised manuscript. This includes the previously cited value for mKikGR of 0.75%, which comes close to the values estimated for PA-SiR-Halo on HaloTag (0.86% when activated at 340 nm).

5) I am not keen on microtubule diameter as a measure, but if given I think it's worth giving the context of how big the microtubule itself is expected to be and how big the labelling system is expected to be.

Response: We thank reviewer #1 for this suggestion and adjusted the manuscript accordingly. We also re-analysed the microtubule diameter using a slightly different method and increased the number of tubules analyzed (see Supporting Information of the revised manuscript). In the following we give a brief explanation why we have changed the method of analysis: Previously, we performed the analysis on rendered images. In the rendered images the localizations are convoluted with a Gaussian distribution which can lead to systematic errors in distance measurements. The new method has the advantage of directly using the coordinates of the localizations and is therefore more reliable. The changes in calculations did not affect the conclusions drawn from them. The new values were introduced in the main text and in the captions of Figure 3 as well as in Supplementary Figure 14g. In addition, we completed this section with more detailed information on the diameter of microtubules (25 nm) and estimates for the increases in size due the employed labeling system CEP41-Halo (CEP41 41 kDa, HaloTag7 33 kDa, total 74 kDa, adding around 5 nm on each side). Consequently, the theoretical diameter of the labeled structure is about 35 nm, which close to the value of 38 nm we measured experimentally (page 5, bottom).

6) In supplementary figure 9 the transmission images are almost impossible to distinguish any features – perhaps the minimum/maximum displayed values can be adjusted with some type of normalisation? At the moment the images do not convey information.

Response: We increased the size of the subfigures and optimized the minimum/maximum displayed such that more features become visible. In general, adherent U-2 OS cells (Supplementary Figure 12 a-b) are rather flat, making it difficult to distinguish structural features in transmission images.

7) What is the impact of the UV needed for photoactivation vs. by-products of the dark state when an organic dye is used in terms of phototoxicity? There must be some sort of tradeoff, and at the moment that isn't clear. I don't want to suggest further experiments as I think there is sufficient evaluation and phototoxicity assessments are very challenging, but I think it should be discussed.

Response: We included a short discussion on the use of UV light for photoactivation, including the potential causes of phototoxicity (toxicity of side-products vs. the light itself) (page 6).

Reviewer #2 (Remarks to the Author):

In this work, the author developed a photoactivatable silicon rhodamines featured with π -conjugation converting, which is employed for live-cell super-resolution microscopy. However, the mechanism of a light-dependent isomerization followed by protonation for spectroscopic properties is hard to be proven with limited the structural information of PA-SiR. In order to address a broader audience, solid evidence of structure and clear illustration of mechanism must be presented to clarify that.

Comments:

1. Light-dependent isomerization followed by protonation is critical for this work. However, the only calculated result is not the compelling evidence to prove the structure of "PA-SiR 4" and the isomerization of PA-SiR. The authors should provide solid and clear evidences, including single crystal X-ray diffraction etc., to illustrate the conversion of π -conjugation.

Response: To address the concerns of reviewer #2, we have attempted to obtain crystal structures of compound **4**, PA-SiR as well as the corresponding activated forms. We succeeded in obtaining a single-crystal x-ray structure of compound **4**. This structure was inserted in Figure 1 (the HOMO-LUMO calculations were therefore moved to Supplementary Figure 3 and the modeled structure of PA-SiR was removed). Additional information on the crystals and the structure can be found in Supplementary Table 11. The crystal structure confirmed the previously modeled structure. The exocyclic double bond was found to be shorter (C7-C14 1.344(2)) than the aromatic bonds (e.g. C4-C5 1.380(2)) and much shorter than C6-C7 1.4896(19). This confirms the presence of two separated aromatic systems (cross-conjugated) as suggested by the characteristic UV-Vis spectrum with absorbance maxima below 400 nm. Unfortunately, repeated attempts to crystallize PA-SiR or its activated form **2** were not successful. However, we have ample other experimental evidence to support the proposed protonation of PA-SiR: We demonstrated by deuterium/ LC-MS experiments (Supplementary Figure 4) that PA-SiR is protonated upon UV-light irradiation. The corresponding peak in the LC chromatogram showed absorbance at around 640 nm. This red-shifted absorbance spectrum of compound **2** is identical to that observed for regular SiR derivatives. The structure of compound **3** was confirmed by ^1H NMR and its reversible formation from PA-SiR can most easily be explained by passing through compound **2**. In conclusion, the revised manuscript provides sufficient experimental support for the structural assignment of PA-SiR and compound **2**.

2. In page 4, the authors mentioned that "PA-SiR-Halo attached to HaloTag can be very efficiently activated". But the mechanism is unclear and confusing. More detailed discussion need about the reason why PA-SiR-Halo attached to HaloTag is more stable than the isolated one.

Response: We agree with reviewer #2 that the much higher stability of activated PA-SiR-Halo conjugated to HaloTag relative to all other activated PA-SiR derivatives is remarkable. We believe that the well-known affinity of HaloTag for rhodamines allows to rationalize this extraordinary

stability, as the underlying interactions should also stabilize activated PA-SiR-Halo. Unfortunately, in the absence of structural information on HaloTag labeled with PA-SiR it is impossible to provide a more detailed explanation. We have nevertheless expanded the discussion of this important feature of PA-SiR-Halo in the revised version our manuscript (page 4, line 17).

3. The authors mentioned that “we prepared conjugates of PA-SiR with ligands for protein labeling”. There are a few discussions about conjugation between PA-SiR-Halo and protein. Please provide more details like gel electrophoresis to prove that PA-SiR-Halo has conjugated with HaloTag.

Response: We added images of gels confirming irreversible reaction of PA-SiR-Halo with HaloTag in Supplementary Figure 9. We also included a paragraph explaining the experimental procedure to do so in the Supplementary Information and referred to these additional experiments in the main text of the revised version of the manuscript (page 4, line 14).

Reviewer #3 (Remarks to the Author):

"Photoactivation of silicon rhodamines via a light-induced protonation" by Reymond, Johnsson and co-workers describes a photoinduced mechanism to activate a cell-permeable, far-red fluorophore ideal for super-resolution microscopy. From a serendipitous discovery in an attempted synthesis of a silicon rhodamine with an alkyl chain, a protonated non-fluorescent form was isolated. Interestingly, UV irradiation induced the formation of the fluorescent form having outstanding spectroscopic properties. The behavior of the fluorophore was subsequently investigated in great detail and halo-, actin and SNAP derivatives were prepared. Super-resolution microscopy, single-particle tracking experiments and live-cell microscopy revealed a clearly improved behavior, signal-to-background ratio, high permeability and stability.

Overall, the manuscript describes a highly valuable photoactivatable fluorophore with remarkable spectroscopic properties and distinctly enhanced utility in super-resolution microscopy. It is therefore recommended to accept this paper for publication after revisions taking in consideration the following suggestions and comments.

Main comments:

(1) The first analogue of this new class was found serendipitously discovered: the synthetic scheme of preparation would definitely improve clarity when illustrated to the main article.

Response: We appreciate the suggestion of reviewer #3 but believe that inclusion of a synthetic scheme for PA-SiR in the already quite crowded Figure 1 would distract from the main point of our manuscript, which is the fluorophore itself and its properties. However, we added a synthetic scheme as an additional Supplementary Figure 2. In the caption of this Figure we also discuss the synthesis further (see also the 2nd point raised by reviewer #3).

(2) The yields for PA-SiR (SiRA in some SI schemes) and PA-SiR C3 is 5% and 8%, resp.: can these yields be further improved? An explanation of these results should be provided.

Response: We have not yet systematically tried to improve the synthesis of PA-SiR. The low-yielding step of the synthesis is the last step where one major side-product was formed. In the revised version of the manuscript we included the structure of this side product in Supplementary Figure 2, as well as its characterization in the Chemistry section of the Supplementary Information. In addition, we commented on the low yields and possible strategies for improvements.

(3) What is the analogous behavior of related fluorophores such as fluorescein or rhodamine and was previously reported (e.g. J. Org. Chem. 2013, 78, 1833)?

Response: We are grateful to reviewer #3 for drawing our attention to this report. The properties of these compounds are similar to those reported for the related carbopyronines (Hatchard, C. G.; Parker, C. A., *Proceedings of the Royal Society of London. Series A. Mathematical and Physical Sciences* **1956**, 235, (1203), 518-536.). Succinylfluorescein exists in several tautomeric forms at room temperature in aqueous solution and no photoactivation has been reported. Similarly to the carbopyronines the olefinic form could not be isolated as the tautomeric forms are in equilibrium. We added its structure to Supplementary Figure 3 and included the paper in our list of references.

(4) For the conceptual basis of the new class of fluorophores: what is the broader scope of the photoactivation strategy?

Response: The available data on carbopyronine and fluorescein derivatives (see also Supplementary Figure 3) suggest that the approach is not directly applicable to these classes of fluorophores. However, with the many strategies available to tune the HOMO and LUMO of xanthene derivatives, we believe it should be feasible to generate photoactivatable rhodamine and carbopyronine derivatives. We have expanded our discussion on the broader scope of this photoactivation strategy in the manuscript (page 4).

(5) An isomerization/protonation sequence is mentioned in the manuscript: the isomerization step is however not clear and an intramolecular charge-transfer is suggested based on calculations. Is an isomerization required or is protonation taking place directly upon charge-transfer? Otherwise a mechanism shown in Fig.1a would clearly improve the model.

Response: We thank the reviewer for this comment. Indeed the choice of the term isomerization is misleading and we removed this term throughout the text. While more studies are needed to investigate the mechanism in greater detail (which we acknowledge in the revised version of the main text, page 3), we currently believe it involves charge-transfer, followed by protonation as shown in new Supplementary Figure 3 of the revised manuscript.

Minor comment:

Some schemes in the SI use a different nomenclature for PA-SiRs (SiRA)

Response: We are grateful to reviewer #3 for spotting these inconsistencies, which we have corrected in the revised version of our manuscript.

REVIEWERS' COMMENTS:

Reviewer #1 (Remarks to the Author):

My concerns have been addressed and I think the paper is now suitable for publication. I think it will be an important and useful tool for cell biologists.

Reviewer #2 (Remarks to the Author):

The authors have addressed all suggestions, so I recommend publication

Reviewer #3 (Remarks to the Author):

In the revised version of this manuscript, the authors implemented all major changes suggested by the reviewers. Therefore, it is recommended to accept publication of this paper in Nature Communication.